# Effects of Different Ammonia Concentrations on Pulmonary Microbial Flora, Lung Tissue Mucosal Morphology, Inflammatory Cytokines, and Neurotransmitters of Broilers

**DOI:** 10.3390/ani12030261

**Published:** 2022-01-21

**Authors:** Guangju Wang, Qingxiu Liu, Ying Zhou, Jinghai Feng, Minhong Zhang

**Affiliations:** State Key Laboratory of Animal Nutrition, Institute of Animal Sciences, Chinese Academy of Agricultural Sciences, Beijing 100193, China; 82101185163@caas.cn (G.W.); 18754870532@163.com (Q.L.); 15624955881@163.com (Y.Z.); fjh6289@126.com (J.F.)

**Keywords:** ammonia, pulmonary microflora, inflammatory cytokines, neurotransmitters, lung–brain axis

## Abstract

**Simple Summary:**

Ammonia is the dominant pollutant gas in poultry houses, and it is harmful to broilers, especially in the cold season. Exposure to ammonia leads to damage to the respiratory system of broilers, affects the health of broilers, and reduces production performance. To date, the relationship between lung flora and immune system and brain exposed to ammonia is unclear, and there have been numerous studies on the lung–brain axis in recent years. Therefore, the aim of this study is to explore the effects of ammonia on lung microflora, lung tissue mucosal morphology, inflammatory cytokines, and neurotransmitters. Moreover, exploring these mechanisms can help in the development of strategies that alleviate the negative effects of the performance of ammonia. Our study suggests that the damage caused by ammonia to broiler lungs may be mediated by the lung–brain axis.

**Abstract:**

Atmospheric ammonia is one of the main environmental stressors affecting the performance of broilers. Previous studies demonstrated that high levels of ammonia altered pulmonary microbiota and induced inflammation. Research into the lung–brain axis has been increasing in recent years. However, the molecular mechanisms in pulmonary microbiota altered by ambient ammonia exposure on broilers and the relationship between microflora, inflammation, and neurotransmitters are still unknown. In this study, a total of 264 Arbor Acres commercial meal broilers (21 days old) were divided into 4 treatment groups (0, 15, 25, and 35 ppm group) with 6 replicates of 11 chickens for 21 days. At 7 and 21 D during the trial period, the lung tissue microflora was evaluated by 16S rDNA sequencing, and the content of cytokines (IL-1β, IL-6, and IL-10) and norepinephrine (NE), 5-hydroxytryptamine (5-HT) in lung tissue were measured. Correlation analysis was established among lung tissue microflora diversity, inflammatory cytokines, and neurotransmitters. Results showed that the broilers were not influenced after exposure to 15 ppm ammonia, while underexposure of 25 and 35 ppm ammonia resulted in significant effects on pulmonary microflora, inflammatory cytokines, and neurotransmitters. After exposure to ammonia for 7 and 21 days, both increased the proportion of Proteobacteria phylum and the contents of IL-1β and decreased the content of 5-HT. After exposure to ammonia for 7 days, the increase in Proteobacteria in lung tissue was accompanied by a decrease in 5-HT and an increase in IL-1β. In conclusion, the microflora disturbance caused by the increase in Proteobacteria in lung tissue may be the main cause of the changes in inflammatory cytokines (IL-1β) and neurotransmitters (5-HT), and the damage caused by ammonia to broiler lungs may be mediated by the lung–brain axis.

## 1. Introduction

Ammonia is one of the poultry house by-products that are detrimental to broilers and is produced from uric acid and undigested proteins in manure by aerobic or anaerobic bacteria [1]. The ammonia concentration in the poultry houses should not exceed 25 ppm. However, ammonia concentration commonly exceeds 25 ppm during the cold season, and this phenomenon is prevalent all over the world [2,3]. Several studies reported that 25 ppm ammonia can reduce productive performance, amplify immune response [4], and damage the respiratory system [5], leading to low breast muscle and carcass composition in broilers [6]. When ammonia concentration reaches 50 ppm, it can even affect the eyes, causing cornea damage and blurred vision, leading to difficulties in food foraging, which affect the metabolism and performance of broilers [7].

In recent years, numerous studies have found that brain diseases such as Alzheimer’s disease are always accompanied by lung infections [8,9]. This has led to the lung–brain axis, a relatively narrow field in recent years. Several studies have shown that pulmonary microflora can regulate pulmonary inflammation and affect the nervous system and immune microenvironment through various mechanisms [10,11,12]. There is evidence suggesting that bacteria may have an impact on the host physiology by regulating neurotransmitters such as dopamine, norepinephrine, serotonin, or gamma-aminobutyric acid (GABA) [13]. Microflora affects neural development and behavior by regulating the hippocampal serotonergic system in the early stages [14]. Although the detailed mechanism of the lung–brain connection remains unclear, it is known that the central nervous system (CNS) can be affected by pulmonary flora through immunity, HPA axis, nervous system, and metabolites. Accumulating evidence suggests that respiratory tract microflora can be disturbed by external environmental factors [15,16]. Pathogenic microorganisms are common in lung diseases, such as asthma and cystic fibrosis. They are related to the increase in inflammatory markers in the stable state of disease. Environmental stress can lead to the disorder of the respiratory tract microflora by increasing the number of pathogenic bacteria, promoting an excessive secretion of toxins that increases the permeability of mucosa, resulting in a large production of inflammatory cytokines such as IL-1β and IL-6 by the host immune system [17]. Therefore, we hypothesized that the damage caused by ammonia to broilers may not only be caused by the lungs but also that the lung–brain axis may be involved in the process of lung injury.

Thus, in this study, we investigated the effects of different concentrations of ammonia on lung tissue mucosal morphology, microflora, inflammatory cytokines, and neurotransmitters of broilers by exposure to ammonia for 7 days and 21 days, and we also analyzed the relationship between lung tissue microflora diversity, inflammatory cytokines, and neurotransmitters. Our results aimed to provide reference and recommendations for the study of the lung–brain axis and the therapy of lung diseases such as chronic obstructive pulmonary disease (COPD) and asthma.

## 2. Materials and Methods

### 2.1. Experimental Design

A single factor random design was used in this experiment. A total of 280 1-day-old male broiler chickens (Huadu Co., Ltd., Hebei, China) were housed in cages and a temperature- and humidity-controlled room, with free access to feed and water, then reared to 21-days-old. At the age of 22 days, 264 broilers with similar weight were randomly divided into 4 treatment groups, each comprising 6 replicates single-tier cages with 11 birds (single-tier cage size is 0.80 m length × 0.80 m width × 0.40 m height). The 4 treatment groups were the control group (0 ppm), the 15 ppm, the 25 ppm, and the 35 ppm ammonia groups. From day 22 to day 42, the concentrated NH_3_ was delivered in the four exposure chambers (4.5 m length × 3.0 m width × 2.5 height). Birds in the control group were housed in a chamber without NH_3_ addition, whereas birds in ammonia groups were exposed to different ammonia concentrations. The concentrations of NH_3_ in 4 chambers were monitored with a LumaSense Photoacoustic Field Gas-Monitor INNOVA 1412 (Santa Clara, CA) during the entire experiment. Temperature (21 °C ± 1 °C), relative humidity (60 ± 7%), and airflow were controlled during the exposures to ensure a suitable environment. The manure was removed from the chambers every 3 days to reduce the volatilization of extra NH_3_.

### 2.2. Experimental Diet and Feeding Management

The diet used for age 21 to age 42 was based on a corn-soybean meal diet without antibiotics. The corn-soybean meal basal diet (Table 1) was formulated to meet or exceed the National Research Council requirements for broilers for all nutrients. The broilers were reared with the same diet and free feeding and vaccination against Marek’s disease, Newcastle disease, and bronchitis.

### 2.3. Sample Collection

On days 7 and 21 of the experiment, three broilers of approximately average weight were selected from each replicate for tissue sample collection. After the broilers were sacrificed, the left lung tissues were isolated and washed with PBS. Three lung tissues from each replicate were mixed into one sample (six samples were harvested from each treatment group) and stored in a −80 °C refrigerator for further analysis.

### 2.4. Morphological Observation of Lung Tissue Mucosa and Determination of Serum Immune Cytokines and Neurotransmitters

Pieces of lung tissues from birds of four groups were fixed in 4% paraformaldehyde. Fixed tissues were embedded in paraffin, then sectioned to 3 μm thickness and stained with Mayer’s hematoxylin and eosin. The contents of interleukin (IL-1β, IL-6, IL-10) and noradrenaline (NE), 5-hydroxytryptamine (5-HT) in lung tissue were detected by an enzyme-linked immunosorbent assay (ELISA) kit (Jiancheng, Co., Ltd., Nanjing, China).

### 2.5. DNA Extraction and PCR Amplification 16s rDNA

Microbial DNA was extracted from lung tissue samples using the E.Z.N.A.^®^ soil DNA Kit (Omega Bio-tek, Norcross, GA, USA) and according to the manufacturer’s protocols. Then, DNA quality was assessed using a 1% agarose gel electrophoresis. The V3-V4 hypervariable regions of the bacteria 16S rDNA gene were amplified with primers 338F (5′-ACTCCTACGGGAGGCAGCAG-3′) and 806R (5′-GGACTACHVGGGTWTCTAAT-3′) by thermocycler PCR system (GeneAmp 9700, ABI, Foster City, CA, USA). The PCR reactions were conducted, and the resulted PCR products were extracted from a 2% agarose gel, further purified, and finally, quantified using the QuantiFluor™-ST (Promega, Madison, WI, USA) according to the manufacturer’s protocol.

### 2.6. Illumina MiSeq Sequencing

Purified amplicons were pooled in equimolar and paired-end sequenced (2 × 300) on an Illumina MiSeq platform (Illumina, San Diego, CA, USA) and according to standard protocols provided by Majorbio Bio-Pharm Technology Co., Ltd. (Shanghai, China).

### 2.7. Statistical Analysis

All statistical analyses for factor measurements of the difference between groups were conducted using one-way ANOVA analysis available with the SAS 9.1 software. Differences among means were tested by Duncan’s multiple range test. The data of inflammatory cytokines and neurotransmitters are presented as mean ± SEM. The data on lung tissue microflora were analyzed by a cloud platform (Meiji Co., Ltd., Shanghai, China). The correlation of inflammatory cytokines and neurotransmitters, respectively, with the abundance of lung tissue microbial species was calculated using Spearman’s coefficient. The replicate cage served as the experimental unit, and *p* < 0.05 was considered statistically significant.

## 3. Results

### 3.1. Effect of Ammonia on Lung Tissue Mucosa of Broilers

The effect of ammonia on lung tissue mucosa is shown in Figure 1. After 7 days of ammonia exposure, compared with the control group, inflammatory cell infiltration occurred in the 15 ppm group, local tissue hemorrhage occurred in the 25 ppm group, and a large number of red blood cells and necrotic cell masses were observed in the bronchus in the 35 ppm group. After 21 days of ammonia exposure, compared with the control group, a large number of red blood cells were found in the bronchus in the 15 ppm group, local tissue hemorrhage occurred in both the 25 ppm group and the 35 ppm group, and focal infiltration of inflammatory cells around the smooth muscle bundle of the bronchus. In addition, connective tissue hyperplasia was also found between lung lobules in the 35 ppm group.

### 3.2. Effect of Ammonia on Pulmonary Microflora of Broilers

The effect of ammonia on the pulmonary microflora of broilers is shown in Figure 2. At the phylum level, Firmicutes, unclassified Bacteria, Proteobacteria and Bacteroidetes are the dominant bacterial phyla. Only the proportion of Proteobacteria was significantly different between four groups after 7 and 21 days long exposure (*p* < 0.05). At the genus level, only unclassified_f_Ruminococcaceae was significantly different between the four groups after 7 days long exposure (*p* < 0.05), but no significant difference after 21 days long exposure.

### 3.3. Effect of Ammonia on Inflammatory Cytokines in Lung Tissue of Broilers

The effect of different ammonia concentrations on inflammatory cytokines in lung tissue under exposure to ammonia for 7 days and 21 days is shown in Figure 3. The results show that the level of IL-1β in lung tissue significantly increased under exposure to ammonia for 7 days in the 25 and 35 ppm group compared with the control group (*p* < 0.05), but there was no significant difference in IL-6 and IL-10 contents between the four groups. After 21 days long exposure, the levels of IL-1β and IL-6 in the 35 ppm group were significantly higher than that in the control group (*p* < 0.05), but there was no significant difference compared with the 15 ppm and the 25 ppm group. Compared with the control group, the content of IL-10 significantly increased in the three ammonia groups after 21 days long exposure (*p* < 0.05).

### 3.4. Effect of Ammonia on Neurotransmitters in Lung Tissue of Broilers

The effect of different ammonia concentrations on neurotransmitters in lung tissue after 7 and 21 days long exposure is shown in Figure 4. The results show that there was no significant difference in NE content between the four groups after 7 and 21 days long exposure. The content of 5-HT decreased with the increase in ammonia concentration. Compared with the control group, the content of 5-HT significantly decreased in the 35 ppm group after 7 and 21 days long exposure (*p* < 0.05).

### 3.5. Correlation Analysis of Pulmonary Microflora with IL-1β, IL-6, IL-10, and NE, 5-HT

The correlations between the top 10 most abundant phylum, inflammatory cytokines, and neurotransmitters in lung tissue were analyzed (Figure 5). The results show that the proportion of Proteobacteria was negatively correlated with 5-HT (*p* = 0.031) and positively correlated with IL-1β (*p* = 0.002) after 7 days long exposure. However, after 21 days long exposure, there were no significant correlations between pulmonary microflora, inflammatory cytokines, and neurotransmitters.

## 4. Discussion

In poultry houses, atmospheric ammonia has been linked to damage to the respiratory tract mucosa and a reduction in resistance to respiratory diseases [18]. Numerous studies have found that ammonia concentration, once exceeding 25 ppm, may have adverse effects on poultry health and production [19,20]. In the current study, we observed that with an increase in ammonia concentration, lung tissue showed different degrees of damage. A previous study reported that broilers exposed to 20 ppm of ammonia over a period of 42 days indicated pulmonary edema, congestion, and hemorrhage [21]. Another study found that exposure to 25 ppm of ammonia resulted in a decrease in performance by impairing the immune response [5]. In this study, we found that the microflora of lung tissue would be disturbed in broiler chickens exposed to 35 ppm of ammonia, the level of inflammatory cytokines increased, and the level of neurotransmitters decreased. These results indicate that exposure to 35 ppm ammonia concentration would have a detrimental impact on broilers’ health.

The respiratory tract of animals is connected to the external environment and, therefore, be widely affected by external factors. The airway microbial that colonizes the mucosa may be the first to be affected by external factors [15]. Germ-free mice exhibited a lack of normally developed immune system and mucosal alternations, both of which can be restored by colonizing microbiota [22,23]. Therefore, the lung injury in the ammonia group may be due to the disorder of microflora in lung tissue. In the current study, we found that the dominant flora of lung tissue includes Firmicutes, Proteobacteria, and Bacteroidetes. These results are consistent with previous reports that showed the existence of the phyla Proteobacteria, Firmicutes, Tenericutes, Actinobacteria, Bacteroidetes, and Chlamydia/Verrucomicrobia in the respiratory tract of domestic and wild birds [24]. A recent study found that air pollutants can cause airway microflora disorders in rats, leading to the colonization of pathogenic bacteria, making the body susceptible to infections [25]. In addition, we found that exposure to ammonia causes lung tissue microflora disorders, and the proportion of Proteobacteria was significantly increased under exposure to ammonia. Shin et al. considered that an increased abundance of Proteobacteria is a potential diagnostic signature of dysbiosis and diseases [26]. Specifically, ammonia, as an environmental pollutant, endangers the homeostasis of respiratory microorganisms in broilers, increasing pathogenic bacteria, which is detrimental to respiratory health.

Cytokines are known to regulate immune and inflammatory responses. It has been suggested that foreign stress can influence the function of immune cells and cytokines [27]. In the current study, the level of inflammatory cytokines IL-1β, IL-6, and IL-10 in lung tissues were affected by ammonia. A previous study reported that exposure to 25 ppm of ammonia could affect the level of cytokines in serum and tracheal tissue of broilers [28]. Further, the level of IL-1β increased in broilers that were exposed to ammonia for 3 weeks, and these released cytokines led to inflammatory responses and multiple organ damage [29]. Our results show an increase in the levels of IL-1β, IL-6, and IL-10 in the 25 ppm and 35 ppm groups compared with the control group. This demonstrates the occurrence of lung inflammation in broilers following exposure to high concentrations of ammonia. Similarly, our last experiment also demonstrates that this exposure to high concentrations of ammonia causes inflammatory damage to broilers’ lung tissues [30].

Neurotransmitters are mainly secreted and released by neurons and are the medium of communication between neurons. They are often used as biomarkers that indirectly reflect the activities and disease status of the central nervous system [31]. One study found that benzo(a)pyrene exposure decreases the content of 5-HT in the urine of 3 to 5-year-old children [32]. At present, there is no study on the effect of ammonia on the content of neurotransmitters in broilers’ respiratory tracts. In the current study, we found that 5-HT in lung tissue significantly decreases following exposure to ammonia. 5-HT can promote the contraction and relaxation of bronchioles and bronchi [33]. Therefore, the decrease in 5-HT content in lung tissue may be due to the decrease in bronchoconstriction and relaxation capacity caused by the continuous inhalation of ammonia. Although due to the presence of the blood-brain barrier, 5-HT in peripheral tissues cannot directly enter the CNS, and studies have shown that peripheral 5-HT can enter the brain through immune cells as a carrier [34]. Furthermore, it has been suggested that the change in 5-HT in the brain was also associated with panic disorder and depression [35]. In addition, previous literature has shown that the concentration of 5-HT in the periphery seemed to have an opposite pattern to that of 5-HT in CNS [34], indicating that the 5-HT concentration of CNS was affected by the periphery 5-HT level. Therefore, when lung inflammation occurs, the changes of 5-HT may be related to CNS through neural and immune pathways.

The existence of respiratory microbiota on the respiratory mucosa probably acts as a gatekeeper that provides resistance to the colonization of respiratory pathogens. The respiratory microbes might be involved in the maintenance of the homeostasis of respiratory physiology and immunity [15]. Previous studies demonstrated that Haemophilus influenzae leads to airway inflammation deterioration, induced by cigarette smoke exposure in COPD (chronic obstructive pulmonary disease) mice [36], and the production of the pro-inflammatory mediators interleukin-6 (IL-6) and IL-1β [37]. This indicates that there is a close relationship between respiratory tract microflora and inflammatory cytokines. In this study, Spearman’s correlation analysis showed that the presence of Proteobacteria positively correlates with IL-1β and negatively correlates with 5-HT when the broilers were exposed to ammonia for 7 days. The increase in Proteobacteria has been proved to be positively correlated with intestinal inflammation [38]. Moreover, a comparison of the bacterial composition of patients with or without asthma demonstrates, in different studies, a higher abundance of Proteobacteria in asthmatic patients. It is generally known that an increase in Proteobacteria represents an imbalance of microflora. 5-HT may either be pro- or anti-inflammatory in monocytes. In the current study, lung function was impaired after ammonia exposure, and the decrease in 5-HT level meant that 5-HT might be an anti-inflammatory factor in lung tissue. Therefore, it is proven that the microflora disturbance caused by an increase in the Proteobacteria population under ammonia exposure may be the main cause of IL-1β content’s increase and 5-HT level’s decrease.

There is growing evidence that the lung and brain are an interrelated system—if one of them is damaged, the other will also be affected and vice versa. The relationship between them is mediated through inflammatory, neurological, and endocrine signaling pathways [39]. Pulmonary microbiota can regulate pulmonary inflammation, which is correlated with cytokines (IL-6, IL-10, et al.). In addition, the microflora can also be connected to the brain through signal molecules such as neurotransmitters (5-HT, NE, etc.). In 2013, Clarke and colleagues reported that the early-life microbiome regulates the hippocampal serotonergic system in a sex-dependent manner [40]. However, some studies have shown that there is a link between serotonin in the lungs and CNS. In addition, serotonin from the lungs can enter CNS through the carrier. Therefore, 5-HT may transmit signals of lung inflammation to the brain. In general, lung microorganisms are disturbed by ammonia, which leads to lung inflammation and lung tissue and mucosal damage; these abnormal changes may be transmitted to the CNS by 5-HT as a signal molecule. This may be one of the lung–brain axis regulation pathways, and the mechanism needs to be explored by further investigation.

## 5. Conclusions

In conclusion, 15 ppm air ammonia barely affects broilers, while 25 ppm and 35 ppm disturbs the pulmonary microflora of group broilers, affecting the immune system and 5-HT level of 35 ppm group broilers. Overall, exposure to 35 ppm ammonia had the most severe effect on broilers. Moreover, under ammonia exposure for 7 days, the pulmonary microflora disturbance caused by an increase in Proteobacteria may lead to an increase in IL-1β and a decrease in 5-HT. Furthermore, this change may be mediated by one of the pathways in the lung–brain axis. These findings will provide a reference and new recommendations for the study of the lung–brain axis and the therapy of lung diseases, such as COPD and asthma. Although our understanding of the lung–brain axis is still very limited, research in recent years has yielded greater progress. Nevertheless, this paper suggests that the damage caused by ammonia to broiler lungs may be mediated by the lung–brain axis.

## Figures and Tables

**Figure 1 animals-12-00261-f001:**
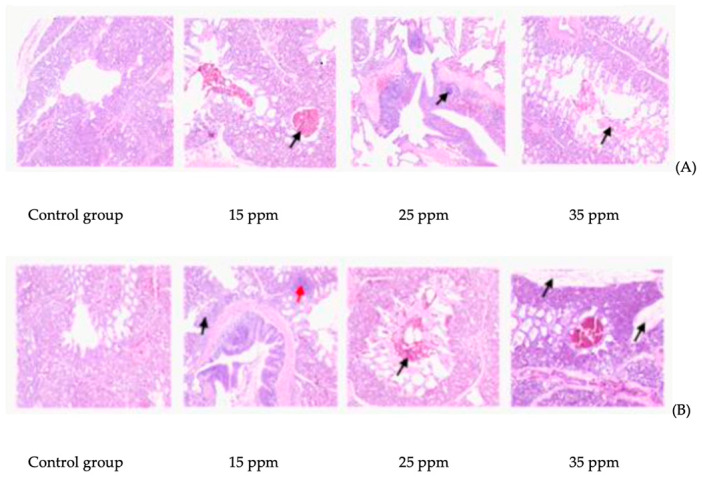
Effects of different ammonia concentrations on lung tissue mucosa under exposures to 7 d (**A**) and 21 d (**B**) ammonia.

**Figure 2 animals-12-00261-f002:**
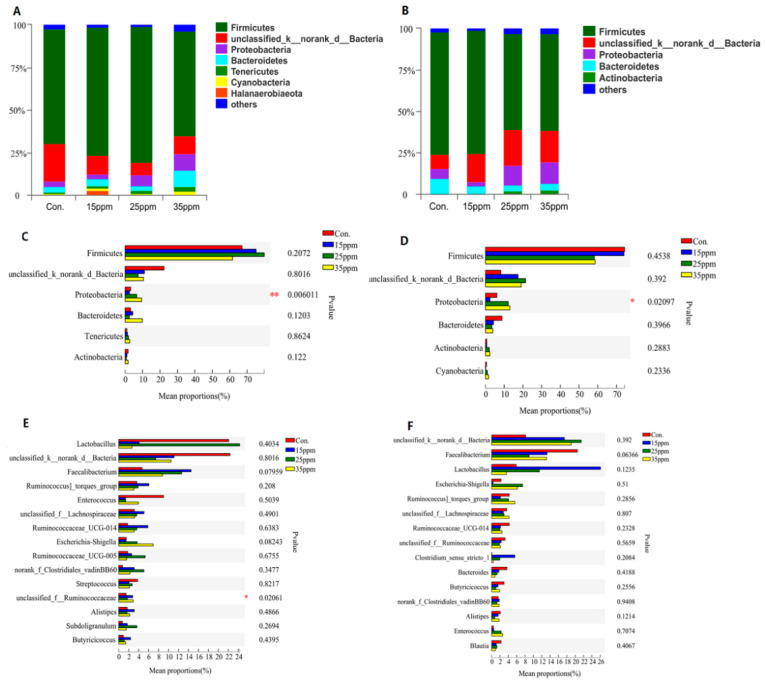
Accumulation map of lung tissue microorganism abundance at the phylum level under exposures to 7 d (**A**) and 21 d (**B**) ammonia. Only the abundant phyla are presented, and other phyla were pooled into ‘Others.’ Difference of student’s *t*-test bar plot on phylum level between four groups under exposures to 7 d (**C**) and 21 d (**D**) ammonia and only the top six abundant phyla are presented. Difference of student’s t-test bar plot on genus level between four groups under exposures to 7 d (**E**) and 21 d (**F**) ammonia and only the top 15 abundant genera are presented. * represents *p* < 0.05 and ** represents *p* < 0.01.

**Figure 3 animals-12-00261-f003:**
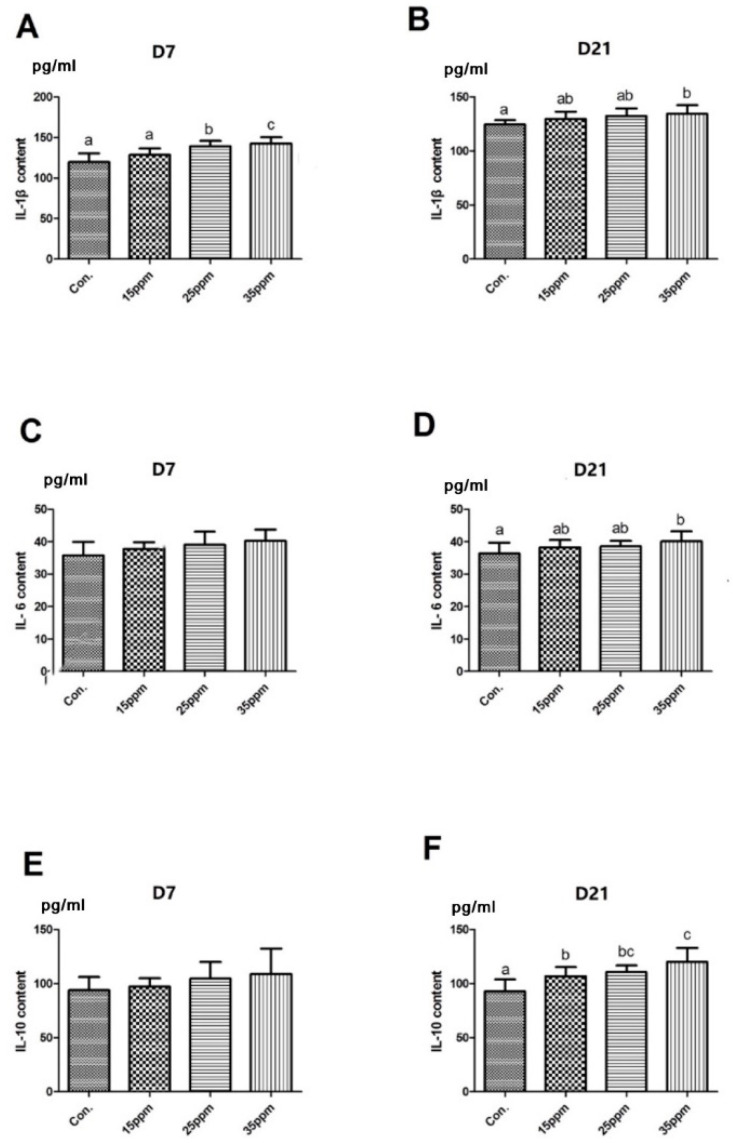
Effect of different ammonia concentrations on IL-1β in lung tissue under exposures to 7 d (**A**) and 21 d (**B**) ammonia. Effect of different ammonia concentrations on IL-6 in lung tissue under exposures to 7 d (**C**) and 21 d (**D**) ammonia. Effect of different ammonia concentrations on IL-10 in lung tissue under exposures to 7 d (**E**) and 21 d (**F**) ammonia. Means with different letters are significantly different at *p* < 0.05.

**Figure 4 animals-12-00261-f004:**
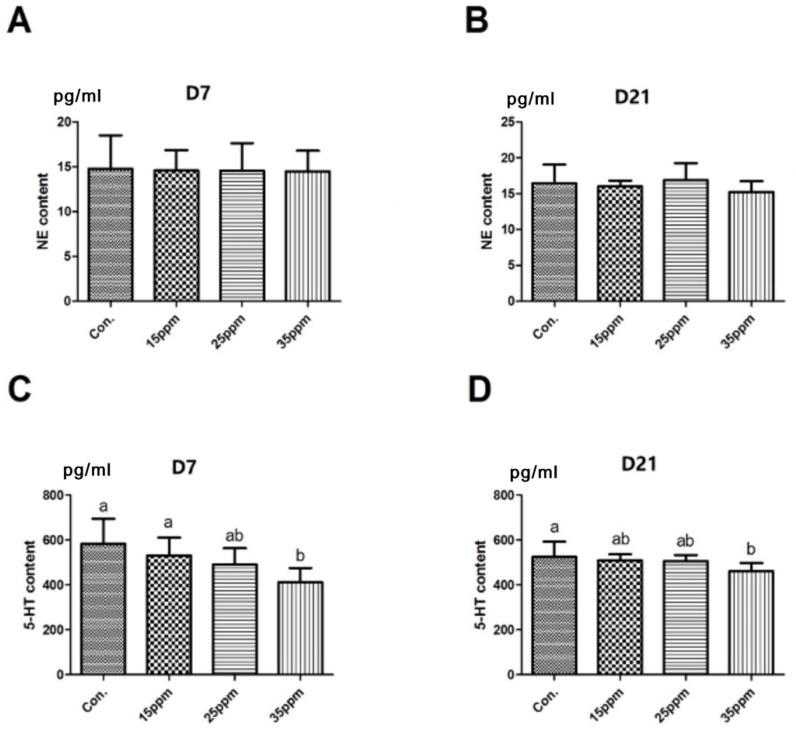
Effect of different ammonia concentrations on NE in lung tissue under exposures to 7 d (**A**) and 21 d (**B**) ammonia. Effect of different ammonia concentrations on 5-HT in lung tissue under exposures to 7 d (**C**) and 21 d (**D**) ammonia. Means with different letters are significantly different at *p* < 0.05.

**Figure 5 animals-12-00261-f005:**
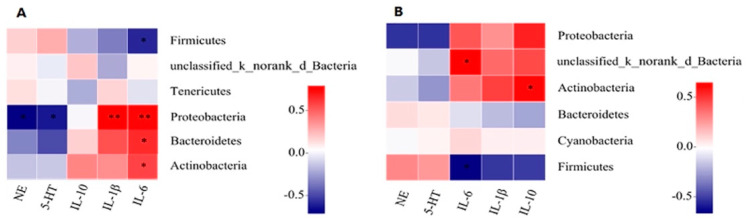
Correlated analysis of the top 10 most abundant phylum with IL-1β, IL-6, IL-10, and NE, 5-HT in lung tissue under exposures to 7 d (**A**) and 21 d (**B**) ammonia. Only the top 10 abundant phyla are presented; * represents *p* < 0.05 and ** represents *p* < 0.01. A correlation coefficient greater than 0 indicates a positive correlation, and less than 0 indicates a negative correlation.

**Table 1 animals-12-00261-t001:** Composition and nutrient levels of the basal diet (as-fed basis).

Items	Content (%)	Items	Content (%)
Ingredients		Nutrient levels ^(2)^	
Corn	56.51	ME/(MJ/Kg)	12.73
Soybean meal	35.52	Crude protein	20.07
Soybean oil	4.50	Ca	0.90
NaCl	0.30	Available phosphorus	0.40
Limestone	1.00	Lys	1.00
CaHPO_4_	1.78	Met	0.42
d L-Met	0.11	Met + Cys	0.78
Premix ^(1)^	0.28		
Total	100.00		

^(1)^ Premix provided the following per kg of the diet: VA, vitamin A 10,000 IU; VD3, vitamin D_3_ 3400 IU; VE, vitamin E 16 IU; VK3, vitamin K3 2.0 mg; VB1, vitamin B_1_ 2.0 mg; VB2, vitamin B_2_ 6.4 mg; VB6, vitamin B_6_ 2.0 mg; VB12, vitamin B_12_ 0.012 mg; pantothenic acid calcium 10 mg; nicotinic acid 26 mg; folic acid 1 mg; biotin 0.1 mg; choline 500 mg; Zn (ZnSO_4_·7H_2_O) 40 mg; Fe (FeSO_4_·7H_2_O) 80 mg; Cu (CuSO_4_·5H_2_O) 8 mg; Mn (MnSO_4_·H_2_O) 80 mg; I (KI) 0.35 mg; Se (Na_2_SeO_3_) 0.15 mg. ^(2)^ Calculated values.

## Data Availability

Not applicable.

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
