# Peer review of "Effects of Different Ammonia Concentrations on Pulmonary Microbial Flora, Lung Tissue Mucosal Morphology, Inflammatory Cytokines, and Neurotransmitters of Broilers"

_animals, 2022, doi:10.3390/ani12030261_

Round 1

Author Response

Dear Reviewer,

Thank you for your letter and for the reviewer’s comments concerning our manuscript entitled “Effects of different ammonia concentrations on pulmonary microbial flora, Lung tissue morphology, inflammatory cytokines and neurotransmitters of broilers”(animals-1537604). Those comments are valuable and very helpful for revising and improving our paper, as well as the important guiding significance to our researches. We have studied comments carefully and have made correction which we hope meet with approval. Revised portion are marked in red in the paper.

The main corrections in the paper and the responds to the reviewers comments are following:

Materials and methods

The arrangement of the animal trial is not clear. What does it mean in line 84 “6 replicated single-tier cage with 11 replicates”? Probably 11 birds in one cage, but it isn’t a single cage.

Response: Thank you for your correction, we have adjusted the expression about the replicates of four groups. Please see line 112.

More detailed description needed on the size of the chambers where the ammonia exposure was applied (size of chambers, air change, source of ammonia etc.). Why only the control group was housed in a separate chamber (line 88)?

Response: Thanks, we have applied the chamber size and the mode of ammonia delivery. Besides, we just want to emphasize the control group will not be affected by ammonia, and we have made some adjustments of expression. Please see line 115-116.

The measured nutrient content of the experimental diets should also be given (Table 1.)

Response: Thanks, the measured nutrient content of the experimental diets is configured in accordance with the NRC requirements with negligible errors.

Avoid abbreviations like AP, CP if it is not written elsewhere.

Response: Thanks, we have added the full name in the basic diet instead of abbreviations. Please see Table 1.

The routine vaccination program should be given. Vaccination against bronchitis could be relevant.

Response: Thanks, we have added the routine vaccination program. Please see line 128-129.

What was the final preparation of the frozen lung samples before the measurement of bacteria, cytokines and neurotransmitters? Probably separate aseptic samples would have been needed for the next general sequencing.

Response: Since we have collected enough samples and only a small number of samples are needed to determine each index, we only need to separate the required mass from a tissue sample to determine the index.

It needs explanation why the measured cytokines and neurotransmitters have been used. What are differences in their functions in the chicken.

Response: As far as we know, IL-1β is considered to be a typical multifunctional cytokine, affecting almost all types of cells, whether acting alone or in combination with other cytokines. IL-1β is essential for cellular defense and tissue repair in almost all tissues and is associated with pain, inflammation, and autoimmunity. IL-1β is also involved in neuroprotection, tissue remodeling, and repair.

IL-6 is a pro-inflammatory factor and is an important medium for acute-phase reactions. Besides, it is able to cross the blood-brain barrier and begin to synthesize PGE2 in the hypothalamus.

IL-10 is an anti-inflammatory factor, Il-10 is widely used in pulmonary disease research. For example, IL-10 plays an important regulatory role in human allergic diseases. Studies have shown that the expression of IL-10 in the respiratory tract of healthy subjects is higher than that in patients with asthma and allergic rhinitis. We selected inflammatory factors with different functions in order to confirm lung inflammation. In addition, many studies have proved that IL-1β and IL-10 are related to microorganisms, and intestinal microbiota can induce the secretion of inflammatory factors and affect intestinal development.

NE and 5-HT are important neurotransmitters and play a role in the brain-gut axis. 5-HT is a monoamine neurotransmitter that modulates nerve development, airway contractility, and immune response. Serotonin receptors 5-HT2A and 5-HT4 have roles in neuronal excitation, smooth muscle contraction, intracellular calcium modulation, and cytokine release. It has been reported that intestinal microorganisms are connected to CNS through 5-HT, which is part of the brain-gut axis. Therefore, we focused on measuring 5-HT in the lungs to explore whether 5-HT plays a role in the lung-brain axis.

In the statistics, the method of the correlation analysis is missing. Explanation needed why the correlation between cytokines and neurotransmitters have not been studied.

Response: Thanks, we have added the methods of correlation analysis in statistical analysis. Please see line 176-177

To the best of our knowledge, studies linking cytokines and neurotransmitters are very limited, especially in the field of animal science.

The evaluation could be also carried out as a two-way ANOVA with the length of the exposure and the ammonia concentration as main factors.

Response: Thanks for your suggestion, It may be applied in our subsequent trials.

Results

The lung tissue morphology is described as mucosal results. They were not only mucosal samples, but also connective tissue and lung lobules.

Response: Thanks, we have replaced the expression about the lung tissue morphology.

Fig 1 contains only some typical microscopic pictures of each treatment. Beside this, more information needed on the frequency of theses disorders in the other replicate animals.

Response: We think the results of lung tissue mucosal morphology is hard to classify because the degree of damage spans a wide range.

Regarding the composition of bacteriota, why only the changes on phyle level are showed? What about the other taxonomic levels?

Response: We apologize for the presentation of our charts which causes obstacles to your understanding. In Figure 2, C and D are about the changes at the phyla level, while E and F are about the changes at the genus level.

Discussion

References missing in line 219 (“Numerous studies have found..”)

Response: We are sorry about the missing reference and we have added that.

In the discussion it is confusing, that results of birds and mammals are not separated. The immune system of birds is different from those of mammals, which should be highlighted. The other disturbing part is that the cited literature contains sometimes the total immune responses of the animals, including also the gut associated immune system. It should be separated and focusing only on the lung related responses.Beside the correlation between the different parameters, the mechanism behind these interactions should also be explained in more details.

Response: At present, there are few studies on the effects of ammonia on the lung flora, immune system, and nervous system of poultry, so we cited some studies in mammals such as mice and humans when discussing the effects of ammonia on lung microflora, neurotransmitters and inflammatory factors. Nevertheless, we still have some changes in the expression.

At last, we have revised the whole manuscript according to the problems you mentioned in the additional remarks.

We tried our best to improve the manuscript and made some changes in the manuscript. These changes will not influence the content and framework of the paper. And here we did not list the changes but marked in red in revised paper.
We appreciate for Editors/Reviewers'warm work earnestly, and hope that the correction will meet with approval
Once again, thank you very much for your comments and suggestion.

Reviewer 2 Report

Minor comments:

  1. The abstract has no background.
  2. Language correction of style is much needed e.g., lines 25-28 etc.
  3. Lines 87-89 – the sentence has no sense – computer programmed – what???
  4. How were animals sacrificed?
  5. Description (lines 108-112) is elusive and must be improved/corrected.
  6. Line 115 – both groups??? What does it mean? There are 4 groups as described by the authors.
  7. “…through conventional histological procedures…” there is no such thing. Please provide a brief description or relevant reference.
  8. Were the tissues embedded in paraffin or resin? This is also not explained according to the thickness of the section.
  9. ANOVA? Which ANOVA? Please provide the used model.
  10. Figure 1. Dramatically low quality and resolution. There is no way to distinguish the changes described by the authors. The pictures are also too small. Figure captions are useless. No scale bars. No description of what the arrows point to.
  11. Who evaluated the histology/histopathology???
  12. Figure 3 and 4. Whiskers means? The legend should be omitted because the bars and x-axis indicate the groups sufficiently. There are no scale units on the y-axis.

Author Response

Dear Reviewer,

    Thank you for your letter and for the reviewer’s comments concerning our manuscript entitled “Effects of different ammonia concentrations on pulmonary microbial flora, Lung tissue morphology, inflammatory cytokines and neurotransmitters of broilers”(animals-1537604). Those comments are valuable and very helpful for revising and improving our paper, as well as the important guiding significance to our researches. We have studied comments carefully and have made correction which we hope meet with approval. Revised portion are marked in red in the paper.

The main corrections in the paper and the responds to the reviewers comments are following:

The abstract has no background.

Response: Thanks for your reminder, we have added the background in the abstract. Please see line 19-24 .

Language correction of style is much needed e.g., lines 25-28 etc.

Response: We have corrected the style of language. Please see line 29-32.

Lines 87-89 – the sentence has no sense – computer programmed – what???

Response: We have deleted the “computer programmed” because that is superfluous. please see line 121.

How were animals sacrificed?

Response: Birds were euthanized by manual cervical dislocation and then exsanguinated for tissue sampling. 

Description (lines 108-112) is elusive and must be improved/corrected.

Response: We have corrected the sentence that you mentioned. Please see line 118-119.

Line 115 – both groups??? What does it mean? There are 4 groups as described by the authors.

Response: We made a mistake in expression due to the limited English proficiency. So we have replaced“both groups” with “four groups”.

“…through conventional histological procedures…” there is no such thing. Please provide a brief description or relevant reference.

Response: We have revised the description of the methods of morphological observation. Please see line 156-157.

Were the tissues embedded in paraffin or resin? This is also not explained according to the thickness of the section.

Response: The lung tissues were embedded in paraffin, and we have updated the expression. Please see line 157.

ANOVA? Which ANOVA? Please provide the used model.

Response: I'm sorry we left out the description of ANOVA. We used one-way ANOVA.

Figure 1. Dramatically low quality and resolution. There is no way to distinguish the changes described by the authors. The pictures are also too small. Figure captions are useless. No scale bars. No description of what the arrows point to.

Response: We have adjusted the figures.

Who evaluated the histology/histopathology???

Response: The histopathology was evaluated by Qingxiu Liu.

Figure 3 and 4. Whiskers means? The legend should be omitted because the bars and x-axis indicate the groups sufficiently. There are no scale units on the y-axis.

Response: The whiskers of Figure 3 and 4 represents the maximum and minimum values between different samples of each indicator, and we have adjusted the legends.

We tried our best to improve the manuscript and made some changes in the manuscript. These changes will not influence the content and framework of the paper. And here we did not list the changes but marked in red in revised paper
We appreciate for Editors/Reviewers'warm work earnestly and hope that the correction will meet with approval.
Once again, thank you very much for your comments and suggestion.

Round 2

Author Response

Dear Reviewer,

    Thank you for your comments concerning our manuscript entitled “Effects of different ammonia concentrations on pulmonary microbial flora, lung tissue morphology, inflammatory cytokines and neurotransmitters of broilers”(animals-1537604).

Regrading Table 1., in the future I suggest to use the breeder’s recommendation instead of NRC 1994. The broilers today have much higher requirements. But since the main focus of the trial was not nutrition, it can be accepted.

Response: Thank you for your tolerance and advice on our experimental diets. We will pay attention to this issue when we conduct the next experiment.

According to your answer the sampling was not aseptic. How was the environmental microflora and the microflora of the other tissue samples excluded before the microbial analysis?

Response: We are very sorry we didn't answer your question completely. We described only the treatments for measuring cytokines and neurotransmitters.

In fact, the operation was kept aseptic when conducting microbiological analysis.

Please allow us to re-answer your question as follows:

1. The whole process was operated in a fume hood, sterilized by ultraviolet and wiped with alcohol to ensure that samples are protected from environmental microorganisms.
2. Wearing aseptic gloves and disposable sample spoons when loading samples.
3. All prepared consumables are aseptic.
4. The amplification of negative control ck, prepared under the same conditions from adding samples to PCR is based on the amplification of no-template by control PCR.

Thanks for the detailed description of the cytokine and neurotransmitter functions. It should be part of the introduction.

Response: We have included some descriptions of cytokines and neuropeptides in the introduction for completeness.

Special thanks to you for your good comments. 

Kind regards,

Guangju Wang.